# Single-Cell Analysis of Bone-Marrow-Disseminated Tumour Cells

**DOI:** 10.3390/diagnostics14192172

**Published:** 2024-09-29

**Authors:** Kevin Wang Leong So, Zezhuo Su, Jason Pui Yin Cheung, Siu-Wai Choi

**Affiliations:** Department of Orthopaedics and Traumatology, School of Clinical Medicine, Faculty of Medicine, The University of Hong Kong, Hong Kong, China; kevinswl@connect.hku.hk (K.W.L.S.); zezhuo@connect.hku.hk (Z.S.); cheungjp@hku.hk (J.P.Y.C.)

**Keywords:** bone metastasis, bone marrow, disseminated tumour cells, metastastic mechanisms

## Abstract

Metastasis frequently targets bones, where cancer cells from the primary tumour migrate to the bone marrow, initiating new tumour growth. Not only is bone the most common site for metastasis, but it also often marks the first site of metastatic recurrence. Despite causing over 90% of cancer-related deaths, effective treatments for bone metastasis are lacking, with current approaches mainly focusing on palliative care. Circulating tumour cells (CTCs) are pivotal in metastasis, originating from primary tumours and circulating in the bloodstream. They facilitate metastasis through molecular interactions with the bone marrow environment, involving direct cell-to-cell contacts and signalling molecules. CTCs infiltrate the bone marrow, transforming into disseminated tumour cells (DTCs). While some DTCs remain dormant, others become activated, leading to metastatic growth. The presence of DTCs in the bone marrow strongly correlates with future bone and visceral metastases. Research on CTCs in peripheral blood has shed light on their release mechanisms, yet investigations into bone marrow DTCs have been limited. Challenges include the invasiveness of bone marrow aspiration and the rarity of DTCs, complicating their isolation. However, advancements in single-cell analysis have facilitated insights into these elusive cells. This review will summarize recent advancements in understanding bone marrow DTCs using single-cell analysis techniques.

## 1. Introduction

Bone metastasis occurs when cancer cells spread from the primary tumour to the bone marrow, forming new tumours in the bone. Among all types of metastatic sites, the bone is one of the most common sites of metastasis [1,2,3]. Clinical studies have also found that bones are the most common sites of first metastatic relapse [1,4]. Bone metastasis is common in patients with prostate (up to 85% metastasize to the bone), breast (70%), and lung (40%) cancers [1], and these cancer types account for 80% of all bone metastasis cases [5]. Breast cancer frequently metastasizes and affects the spine, rib, pelvis, proximal humeri, and femora [6]. Though tumour metastasis is responsible for over 90% of cancer-related deaths [7], there is currently no effective therapy for bone metastasis, and treatment is often palliative [8].

Metastasis involves four processes. Cells from the solid tumour break away and intravasate through the blood vessels and enter the circulation. The cells will travel to a distant site and then extravasate and colonize the new site, resulting in a metastasis [5,9,10]. Circulating tumour cells (CTCs) are tumour cells that leave the primary solid tumour and shed into the circulation [11]. CTCs travel through the bloodstream to distant sites, attracted to and trapped by adhesion molecules expressed on the surface of endothelial cells in the bone marrow. Molecular crosstalk between tumour cells and the bone marrow micro-environment is mediated by direct cell-to-cell interactions and paracrine signalling molecules, such as transforming growth factor β (TGFβ), fibroblast growth factors (FGFs), platelet-derived growth factor (PDGF), insulin-like growth factor (IGF), Jagged, parathyroid hormone-related protein (PTHrP), and receptor activator of nuclear factor kappa-Β ligand (RANKL) [10]. CTCs can employ such molecular interactions and invade the bone marrow, thereby becoming disseminated tumour cells (DTCs). Some DTCs survive and remain dormant in the bone marrow for a period of time [12,13], but upon activation, a portion of DTCs will be triggered to form metastases [12]. Evidence supports that the presence of bone marrow DTCs alone, without any detected bone metastasis, is strongly associated with both future bone and visceral metastasis [14,15,16].

There has been much advancement and translational research being conducted in the molecular characterizations and translational research of CTCs in recent years, even though the study of bone marrow DTCs pre-dates CTC investigations [17]. The reasons behind the slow uptake in DTCs are two-fold. Firstly, bone marrow aspiration is an invasive procedure, and therefore the study of it is a lot more challenging. Secondly, the complex architecture of the bone marrow and rarity of DTCs makes the isolation, and therefore the study, of DTCs difficult. However, the recent utilization of single-cell analysis has helped to provide insights into these rare cells. Several tools have been developed to detect, isolate, enrich, and characterize bone marrow DTCs.

Studies characterizing the biology and molecular profiles of bone marrow DTCs are warranted because information regarding the genetic drivers of metastatic potential of cancer, which may not be obtainable by studying CTCs, can be learned. In contrast to CTCs, bone marrow DTCs are usually considered more advanced cancer cells because these cells have already disseminated to distant organs, and previous study has shown that heterogeneity does exist between DTCs and CTCs [18]. Deng et al. (2004) analyzed PIK3CA mutations on CTCs, DTCs, and metastatic tissues from patients with metastatic breast cancer using Sanger sequencing and found that mutational discordance exists among them [18]. In addition, numerous studies have shown that the presence of either CTCs or bone marrow DTCs is associated with poorer clinical outcome in both early and metastatic cancer. Meta-analyses have confirmed that presence of CTCs or bone marrow DTCs in patients with stage I to stage III breast cancer [19,20] or metastatic and early prostate cancer [21] is associated with poor survival and risk of relapse.

In view of the roles of bone marrow DTCs in both bone metastasis and visceral metastasis, establishing their molecular profiles is crucial for developing new therapeutic targets, stratifying metastatic risk, and selecting treatments in patients with cancer. Therefore, we provide here a comprehensive review on the use of single-cell analysis on studying these cells. The technologies used in enrichment, isolation, and characterization of solitary bone marrow DTCs are described. Furthermore, a summary of the contribution of single-cell analysis of bone marrow DTCs on cancer research and its clinical implications is given.

## 2. Single-Cell Analysis of Bone Marrow DTCs

Traditionally, bone marrow DTCs were detected by immunocytochemical (ICC) assays [22]. A number of studies have shown that detection of bone marrow DTCs by ICC assay is associated with poorer survival in patients with early-stage cancers [20,23]. Although the detection of bone marrow DTCs may have significant prognostic value, the biology of bone marrow DTCs has not been well understood because of their rarity and complexity of bone marrow architecture [24]. However, the use of single-cell analysis has helped researchers to go beyond detecting bone marrow DTCs and elucidate their biology. The current methods for enriching, detecting, isolating, and characterizing solitary bone marrow DTCs is described.

### 2.1. DTCs Enrichment

Separation of rare DTCs requires prior enrichment. Although most of the enrichment technologies were primarily developed to capture CTCs from peripheral blood, which is less complex than bone marrow, several techniques have been studied and optimized to enrich DTCs from bone marrow [25,26], using an approach based on the expression of epithelial markers or biophysical properties [27].

Enrichment based on epithelial markers includes positive and negative immunomagnetic selection, namely epithelial cell adhesion molecule (EpCAM) and cytokeratin, and the use of CD45 to remove leukocytes [28,29]. This approach utilizes magnetic beads that have been coated with antibodies tailored to recognize markers found on tumour cells. These beads adhere to DTCs within the sample, and subsequently, a magnet is employed to separate and collect them. The question regarding whether these cells expressing epithelial markers are genuinely tumour cells was addressed through single-cell studies. The majority of these cells were found to exhibit genetic alterations, signifying their status as tumour cells [30,31,32]. However, the dynamic expression of EpCAM during epithelial to mesenchymal transition in bone marrow DTCs may lower the sensitivity of this method [33].

Other enrichment methods that have been developed are based on the density, size, and deformity of DTCs. Cell-density-based methods utilize density gradient centrifugation with OncoQuick or Ficoll to enrich mononuclear cells, including tumour cells [34,35]. Mean recovery rate of tumour cells in bone marrow have been found to be 41% for OncoQuick and 34% for Ficoll, which is similar to recovery in blood [36]. Another approach to capturing DTCs is the microfluidic-based filtration system [26], which is based on tumour cells being less stiff and larger than non-neoplastic cells. The recovery rate and capture rate of microfluidic-based filtration system on bone marrow DTCs were comparable to CTCs in peripheral blood [26] and might perform better than EpCAM-based capture in prostate and non-small cell lung cancer [37,38]. Lastly, the microfiltration technique, which extracts tumour cells from blood cells because of their size differences, has also been evaluated in capturing bone marrow DTCs and has demonstrated an up to 92.1% recovery rate [39].

### 2.2. DTCs Detection and Isolation

Currently, the detection of bone marrow DTCs is mainly based on either immunocytochemistry or reverse transcription polymerase chain reaction (RT-PCR) analysis [34]. Detecting DTCs using immunocytochemistry is achieved by staining specific antigens with immunofluorescence antibodies, followed by morphological analysis [34,40]. Antibodies to epithelial membrane antigen (EMA) was the first staining technique used to identify breast cancer cells in marrow [41]. However, it was no longer recommended as it can yield false positive results by reacting with plasma cells and immature precursor cells in bone marrow [40,42,43]. These results indicated that additional anti-epithelial markers should be used for cancer cells identification. Currently, the most commonly used anti-epithelial markers are anti-cytokeratins, such as A45-B/B3 and AE1/AE3 [40,44]. A study collected bone marrow aspirates from 212 patients with breast or colorectal cancer and 75 patients without cancer and found that cytokeratin-positive cells were only found in the group with cancer [45]. Subsequent studies found cytokeratin-positive cells in bone marrow to indeed be cytogenetically aberrant, justifying the use of anti-cytokeratin antibodies to detect cancer cells [30,46]. In addition to cytokeratin, clinical relevancy of many other markers has been studied in recent years. A study isolated bone marrow DTCs from patients with breast cancer using a pan-cytokeratin marker and either Thomsen nouvelle (Tn) or O-Acetyl-GD3 [47]. A high number of stained DTCs was likely to appear in patients with metastasis, although only limited patients were included in the study. Another study using a combination of antibodies: pan-cytokeratin (A45-B/B3)/chemokine receptor (CXCR4)/transcription factor (JUNB) showed that patients with (CXCR4+JUNB+CK+)-expressing DTCs were at a higher risk for relapse [48]. Other markers being studied, although not exhausted, include mucin-1 (MUC-1), human epidermal growth factor receptor 2 (HER2), CD 176, aldehyde dehydrogenase 1 (ALDH1), urokinase-type plasminogen activator receptor (uPAR) [49,50,51,52,53,54,55]. Nonetheless, no studies have been conducted to compare the clinical relevancy of different antibodies.

One major advantage of immunocytochemical method is that it allows the combination of immunostaining with morphological analysis, which has been proven to improve the specificity of DTCs identification [44]. Consensus has been reached regarding the morphological criteria for identifying DTCs in positive immunostaining cells [56]. In addition, automated digital microscopy and fibre-optic array scanning technology are flow cytometry-based techniques that have been developed to analyze large numbers of positive immunostaining cells [40,57,58]. However, the detection rate of bone marrow DTCs by immunocytochemistry remains variable, and no detection method for bone marrow DTCs has been officially approved [34,59].

To optimize immunofluorescent detection of bone marrow DTCs, researchers have identified three major obstacles to detect bone marrow DTCs by immunocytochemistry [60]. Firstly, bone marrow harbours a large number of immune cells that can bind antibodies, engulf foreign particles, and exhibit higher autofluorescence compared to other cell types [60,61]. Secondly, compared with blood, the bone marrow contains additional proteins, including collagens, that auto-fluoresce [60,62,63]. Lastly, markers utilized for CTC detection may lose specificity for DTCs due to their heterogeneity [60]. As a result, different attempts have been made to improve the performance of immunocytochemistry, such as blocking the Fc receptors on the immune cells’ surface and reducing the fixation time to decrease bone marrow tissue autofluorescence [60].

Besides immunocytochemical methods, real time RT-PCR is another commonly used technique that enables detection of bone marrow DTCs. Since numerous genetic mutations have been found in common solid tumours, detection of bone marrow DTCs mainly relies on the screening of epithelial- or organ-specific mRNA expression not present in normal bone marrow cells [64], including cytokeratin-19, human mammaglobin, and twist-related protein 1 (TWIST1) [65,66]. However, the major disadvantage of using these tissue-specific mRNA species is false positive results due to low transcription in normal bone marrow cells [65,66,67]. In addition, due to the heterogeneity of DTCs, there is no single genetic marker that enables the detection of all bone marrow DTCs in a single patient. Therefore, a multi-marker assay approach is recommended to overcome the aforementioned concerns. The performance of immunocytochemistry and RT-PCR that quantifies cytokeratin-19 and human mammaglobin mRNA expression on detecting bone marrow DTCs are comparable, making RT-PCR a superior method because it is less observer-dependent [64]. It has also been found that a multi-marker RT-qPCR assay, utilizing keratin 19, mammaglobin A, and TWIST1 mRNA markers, identified DTCs in 40% of bone marrow samples, while immunocytochemistry using AE1/AE3 anti-cytokeratin monoclonal antibodies detected DTCs in only 7% of the samples [68]. Furthermore, using nCounter™ system, which detects bone marrow DTCs by detecting and capturing 38 gene targets simultaneously, it was found that six out of eight patients who developed metastatic disease had at least one detectable transcript [65]. Remarkably, in three of these patients, the bone marrow exhibited expression of HER2, even though their primary tumours were HER2-negative, which is a contraindication to trastuzumab therapy. These findings demonstrate that the application of multi-marker detection of gene expression is not only limited to the detection of DTCs but is also useful in predicting early metastasis and identifying a therapeutic target.

Lastly, micro-manipulation or laser microdissection [34] and fluorescence-activated cell-sorting (FACS) instrument are other techniques which allow the sorting and isolation of DTCs [69].

Micromanipulation is the simplest and most convenient way to isolate single cells from intact live tissues. However, it is limited by its low throughput and the requirement of high skilled technicians [70,71]. In contrast, laser microdissection is a technology enabling the isolation of single cells from a fixed live tissue. Cells of interest are first identified by an inverted microscope, and then laser pulse using infrared or ultraviolet light is delivered to capture the targeted cells. Similar with micro-manipulation, laser microdissection is a low-throughput technique. In addition, the use of laser may cause DNA or RNA damage in cells [72,73]. Comparing with micro-manipulation and laser microdissection, FACS is a technique with a very high throughput. In FACS, a cell suspension is prepared, and the targeted cells are tagged with immunofluorescent probes. During cytometry, fluorescence detectors identify cells based on predetermined characteristics. Subsequently, a charge (positive or negative) is assigned to droplets containing the cells of interest, while an electrostatic deflection system guides the charged droplets into designated collection tubes for subsequent analysis. Major limitations of FACS are the requirement of an initial high-quantity cell population and damage of targeted cells during high-flow cytometry [69].

### 2.3. Molecular Characterization of DTCs at the Single-Cell Level

#### 2.3.1. Single-Cell Genomic Analysis

Single-cell genomic analysis has proven useful in understanding the origin and heterogeneity of bone marrow DTCs. However, insufficient DNA content in a single DTC limits the usefulness of high-throughput assays. Thus, a variety of single-cell whole-genome amplification (WGA) kits, based on different WGA techniques, have been developed. Currently, two main WGA techniques, namely ligation-mediated PCR (LM-PCR) and degenerate oligonucleotide-primed PCR (DOP-PCR), have been widely used in the study of bone marrow DTCs [74,75,76]. DOP-PCR utilizes a partially degenerated oligonucleotide primer and a series of cycles with progressively increasing annealing temperatures to ensure uniform yet non-selective amplification of the entire genome. While the exponential amplification of DOP-PCR allows for the amplification of minute initial material, it tends to yield low gene coverage due to uneven amplification patterns resulting from the exponential process [77]. In contrast, LM-PCR utilizes oligonucleotide linkers that ligate to the targeted DNA sequence, followed by the annealing of PCR primers to amplify the targeted sequence [78].

Copy number variation is a well-recognized genomic alteration detected in tumourigenesis [79]. Copy number variation is defined as an increased or decreased number of DNA segments that is equal to or larger than 1kb, while single-nucleotide polymorphisms are defined as genetic alterations in one single base [79,80,81]. Such genetic change has been extensively studied in bone marrow DTCs (Table 1). The first molecular assay used to analyze copy number variations in bone marrow DTCs at a single-cell level is chromosome comparative genomic hybridization (cCGH) [30]. cCGH involves co-hybridizing fluorescently labelled tumour genomic single-stranded DNA and normal reference single-stranded DNA to normal chromosome at metaphase [82]. By analyzing the relative fluorescent intensities along chromosomes, cCGH can detect losses, gains, and amplifications in the tumour genome. However, the resolution of cCGH is limited to alterations of 5–10 Mb [83]. In order to overcome this limitation, Array Comparative Genomic Hybridization (aCGH), a technique combining the principles of CGH and microarrays can be employed [84]. Unlike cCGH, which uses metaphase spreads, aCGH utilizes microarray-based probes containing short oligonucleotides to co-hybridize labelled tumour and normal DNA. Since probes are much smaller than metaphase chromosomes, aCGH allows a much higher-resolution analysis of copy number variations across the entire genome [84].

In the absence of chromosomal copy number variations (as defined by a normal CGH profile), detection of loss of heterozygosity, which includes sub-chromosomal changes, becomes important to delineate whether the isolated cells are malignant or not. Analysis of loss of heterozygosity in bone marrow DTCs is usually conducted by either microsatellite analysis or polymerase chain reaction–restriction fragment length polymorphism (PCR-RFLP).

A single-nucleotide polymorphism (SNP) array is another technology that has been applied in studying bone marrow DTCs. Single-nucleotide polymorphism arrays contain thousands of nucleotide probe sequences which bind to fragmented single-stranded DNA with signal intensity being correlated with the amount of targeted DNA in the sample [97,98].

Finally, Next-Generation Sequencing (NGS), which involves parallel sequencing of billions of DNA strands, has revolutionized single-cell genomic analysis by providing a high-throughput sequencing method to detect different genomics alterations at the single-cell level [99].

#### 2.3.2. Single-Cell Transcriptomic Analysis

Apart from genomic analysis, different techniques have been utilized to study the transcriptome of DTCs at single-cell resolution, namely dot blot hybridization, expression microarray, and single-cell RNA sequencing (sc-RNAseq).

Dot blot hybridization is a technique that can estimate the relative levels of RNA in a sample by first immobilizing nucleic acids of interest onto a membrane, followed by hybridization with complementary sequences, then quantification of RNA of interest [100]. In comparison, expression microarray is a much more powerful technique with much higher throughput because the probes allow for the measurement of the expression levels of thousands of genes simultaneously [101].

Lastly, sc-RNAseq is a powerful technique used to analyze the transcriptome of DTCs, with only one study having utilized this method to study bone marrow DTCs [95]. Single-cell sequencing protocols can be classified into droplet-based protocols (e.g., Drop-seq and Chromium) and plate-based protocols (e.g., SMART-seq2, Cel-seq), which capture cells into microfluidic droplets or wells, respectively [102]. After sequencing and pre-processing of RNA data, various analysis algorithms can be used to infer different information. For example, a large-scale copy number variation inference algorithm, which is based on the principle that large-scale copy number variations can result in up- or downregulation of genes, has been developed [103]. Furthermore, single-nucleotide variation analysis, cell–cell interactions, cell type abundance, differential gene expression, and gene set enrichment analysis are other mechanisms of interests that can be investigated with high-quality RNA data [104].

## 3. Contribution of Bone Marrow DTCs Single-Cell Analysis to the Advancement of Cancer Research

Single-cell analysis revolutionizes the understanding of bone marrow DTCs, unveiling their genetic diversity and impact on cancer progression. It challenges conventional models by revealing early dissemination and genetic divergence from primary tumours. Single-cell studies also highlight the complexity of DTC heterogeneity, shedding light on tumour dormancy and treatment resistance within the bone marrow microenvironment. By identifying unique genetic markers and therapeutic targets, single-cell analysis paves the way for personalized interventions and risk stratification, offering new insights into metastatic cancer management.

### 3.1. Cancer Progression Model

There are two models that explain tumourigenesis and systemic cancer progression: linear progression and parallel progression. The linear model postulates that clones of tumour cells out-compete and obtain all genetic alterations required for dissemination within the primary site under natural selection [105,106]. Therefore, dissemination of tumour cells should be a late event, and bone marrow DTCs should contain most of the mutations present in the primary tumour. However, the observation that patients with non-metastatic disease relapse after curative surgery demonstrate that tumour cells might disseminate early during the disease process. The parallel progression model posits dissemination of tumour cells during the early stage, and thus bone marrow DTCs are exposed to an independent selection pressure and possess a different genetic profile when compared with the primary tumour [105]. As a result, it is unlikely that adjuvant therapies that do not target early genetic events in tumourigenesis will be effective in eradicating disease. Bone marrow DTCs progress independently from the primary tumour, and so it is Impossible to extrapolate data from the primary tumour to DTCs. Though single-cell analysis has played an important role in enhancing our understanding on cancer progression models, a disparity occurs, in that earlier data comparing the genomic profiles of bone marrow DTCs seemed to support parallel progression, but later studies support a linear progression model.

The first single-cell analysis on bone marrow DTCs conducted by Klein (1999) demonstrated that these cells displayed congruent chromosomal aberration profiles with distant metastasis and showed loss of other tumour suppressor genes, namely adenomatous polyposis coli (APC) and cadherin-1 (CDH1) [30]. This landmark study was followed by subsequent studies to trace the origin of bone marrow DTCs by comparing their aberration profiles with the matched primary.

Schmidt-Kittler (2003) found that single-cell cCGH profiles of bone marrow DTCs from breast cancer patients without metastasis displayed significantly less chromosomal aberrations and chromosomal breaks compared with matched primary and bone marrow DTCs from patients with metastasis, suggesting that these cells disseminated early, and further evolution might have been inhibited by a distinct microenvironment [34]. To confirm the hypothesis, Schardt JA (2005) selected all CK+ cells with normal cCGH profiles from the same group of patients and found these cells exhibited a significantly higher loss of heterogeneity when compared with cells from age-matched healthy individuals, which shows that these cells are indeed tumour cells disseminated from breast tissue [31]. Given that over 95% of cells from matched primary tissue displayed chromosomal aberrations detectable by cCGH, it was hypothesized that these cells must be disseminated at a very early stage of disease. These results show that identification of DNA changes that precede dissemination is important to eradicate minimal residual disease. A limitation of the study by Schmidt-Kittler (2003) is that the proliferative potential of disseminated cells was not taken into account; therefore, it is uncertain whether the cells analyzed are capable of metastasis. However, similar results were reproducible on a study by Gangnus (2004), which only collected bone marrow DTCs after evaluating their proliferative potential by short-term culturing [32].

Apart from breast cancer, heterogeneity of chromosomal aberrations were also identified between matched primary and bone marrow DTCs in both metastatic and non-metastatic prostate cancer patients, suggesting that DTCs acquired chromosomal aberrations after dissemination from primary tumours [86]. In addition, heterogenous chromosomal aberrations were found in DTCs from non-metastatic prostate cancer, while DTCs from non-metastatic prostate cancer displayed several characteristic changes, which indicates a selection of advantageous genomic transformations. Using whole-genome amplification and somatic nuclear polymorphism array analysis on DTCs isolated from patients with metastatic prostate cancer, Wu (2016) found that metastatic tissues contained several aberrations that were not detected in the matched DTCs which displayed somatic copy number aberration characteristics specific to prostate cancer [88]. These findings suggested that DTCs might be the precursor of metastatic tissues, which gained further mutations in ectopic sites, supporting the parallel progression model.

To further elucidate the mechanism of early dissemination, Stoecklein (2008) found only bone marrow and lymph node DTCs, but primary tumour cells from operable esophageal cancer exhibited gain in the region of 17q12-21 or HER2 by single-cell cCGH analysis [85].

Stoecklein (2008) isolated bone marrow DTCs, lymph node DTCs, and primary tumour in patients with operable esophageal cancer [85]. Survival analysis showed that patients with HER2 amplified DTCs had significantly poorer survival compared to those without HER2 amplification (median survival difference = 14.6 months, *p* = 0.005). Similarly, a subset of patients with CK+ cells carrying normal cCGH profile in Schardt JA (2005)’s study also exhibited HER2 amplification [31]. These results showed that studying primary tumours alone is by no means adequate, and direct analysis of DTCs might help us to gain insight into the genetic changes and potential therapeutic targets relevant for systemic spread.

Unlike previous studies using low-resolution single-cell cCGH, Mathiesen (2011) developed a high-resolution single-cell aCGH to compare the aberration profiles between bone marrow DTCs and matched primary in patients with early or metastatic breast cancer [75]. It was found that significant aberrations similarities exist between matched primary and bone marrow DTCs, supporting the linear progression model. Similarly, using single-cell next-generation sequencing and SNP-CGH data, Moller (2013) found bone marrow DTCs from a patient with non-metastatic breast cancer showed 99% similarities in copy-number patterns when compared with corresponding primary tumour, indicating that the dominating clone within the primary tumour has outcompeted other clones and gained necessary mutation before dissemination [74].

To explain the discrepancy of results between two schools of thought, Demeulemeester (2016) conducted a study aimed to identify genuine tumour-derived disseminated tumour cells (DTCs) and investigate their genetic profiles in comparison with the primary tumour and lymph node metastasis [87]. The study was conducted on six breast cancer patients, and single-cell sequencing was used to compare the genetic profiles of DTCs with clonal and sub-clonal copy number aberrations profiles and somatic single nucleotide substitution profiles. The results showed that isolated single cells with copy number variation profiles similar to the primary tumour or the lymph node metastasis were classified as genuine tumour-derived DTCs, while other single cells with either neutral copy number variation profiles or distinct copy number variation profiles from the matched primary tumour were labelled as “normal cells” and “aberrant cells of unknown origin”, respectively. Phylogeny reconstruction based on copy number variation profiles showed that genuine DTCs were derived from either the most recent common ancestor (MRCA) or a subclone of MRCA. The study also found that all three patients with genuine DTCs eventually developed distant metastases, while among three patients with aberrant cells of unknown origin, only one showed systemic recurrence. The study concluded that previous studies that identified distinct chromosomal aberrations from matched primary tumours might have been erroneously interpreted to support the parallel progression mode.

### 3.2. Heterogeneity among Bone Marrow DTCs

Extensive intra-tumour heterogeneity exists in primary tumours. This is not a problem in localized disease because local treatments, such as surgery and radiation, can remove the entire tumour. However, whether disseminated tumour cells are genetically heterogenous or homogenous can affect the efficacy of adjuvant systemic treatment, which aims at lowering the risk of metastasis and relapse. Single-cell analysis of bone marrow DTCs has revealed the huge genetic heterogeneity of bone marrow DTCs, which brought into question the use of adjuvant therapies which only rely on a single agent.

Klein (2002) isolated bone marrow CK+ cells from patients with breast, gastrointestinal tract, or prostate cancer [89]. Single-cell cCGH analysis showed that cells from metastatic diseases were significantly more homogeneous than cells from non-metastatic diseases. In addition, cells isolated from metastatic diseases displayed similar genomic changes over time, while cells isolated from non-metastatic diseases were heterogeneous over time. These findings suggest that genetic diversification exists among early-disseminated tumour cells prior to the occurrence of clinical metastasis. These early-disseminated cells mutate in ectopic sites, and some of them will eventually outcompete others, forming a more homogenous cell population. Another study by Czyż et al. (2014) postulated that specific clones within the highly heterogenous DTC population are selected by chemotherapy [90]. Single-cell aCGH microarrays analysis of bone marrow DTCs isolated after different cycles of chemotherapy showed that several clones possessed minimal regions of aberrations shared by primary tumour, DTCs, and distant metastasis. These aberrations may be essential for tumour formation at distant sites and may contribute to chemotherapy resistance.

### 3.3. Tumour Dormancy

The first place in bone where DTCs encounter after leaving the bloodstream is the perivascular niche [107]. Interactions between perivascular niche in bone marrow and DTCs allow DTCs to acquire and maintain stemness traits. Multiple pathways have been found that mediate the interactions. For example, vascular E-selectin and stromal cell-derived factor 1 (SDF1) were identified as critical for DTCs to remain in perivascular niche [108,109]. Interactions between endothelial Jagged 1 (JAG1) with cancer-cell-derived NOTCH1 in the perivascular niche can maintain the stemness of DTCs [110,111]. In addition, mesenchymal stem cells were found to play an important role in transforming breast cancer cells via the WNT-catenin-dependent pathway [112,113]. Finally, the hypoxic state of the bone marrow also promotes DTCs’ stem-like properties and dormancy [114]. As cancer cells with stem cell-like properties are believed to be more dormant than other cancer cells, bone marrow DTCs in the perivascular niche are generally more resistant to chemotherapy, which target cells with active replications [115]. Furthermore, reversibility of stemness in DTCs is a key contributor to secondary metastasis. Bado IL et al. has shown that Zeste homolog 2 (EZH2), a histone methyltransferase, can transiently epigenetically reprogram estrogen-receptor-positive (ER+) breast DTCs in the endosteal niche of bone marrow, driving them toward a stem cell-like and endocrine resistance state [116]. However, once ER+ breast DTCs leave the endosteal niche, their endocrine resistance and stemness are partially recovered, leading to secondary metastatic spread [116].

In addition to contributing to treatment failure, dormant DTCs are notoriously difficult to identify through non-invasive techniques. Single-cell analysis on bone marrow DTCs not only provide valuable evidence supporting tumour dormancy, but it also enables successful identification of pathways relevant to tumour dormancy, which might be therapeutic targets or biomarkers allowing early identification of dormant tumour cells.

Mathiesenet (2011) compared the single-cell aCGH profiles between DTCs from a patient with early breast cancer at the point of diagnosis and three years after diagnosis [75]. They found that the aberration pattern was the same and provided evidence that bone marrow DTCs might stay dormant for several years.

To identify any molecular pathway associated with cancer dormancy, Holcomb (2009) performed aCGH on bone marrow DTCs isolated from patients with localized disease and metastatic disease [91]. Comparison of aCGH profile showed only DTCs from metastatic diseases possessed deviations related to proliferation, supporting the dormancy theory. Notably, only DTCs from metastatic diseases possessed 8q gain, which was associated with metastatic disease in the organ-confined stage [117]. Chéry (2014) performed single-cell transcriptomic analysis on bone marrow DTCs from prostate cancer patients with no evidence of disease or with advanced disease by microarray [92]. Gene-set enrichment analysis showed that p38 pathway was the top differentiating pathway between two groups. The p38 pathway has also been proposed to play an important role in regulating tumour dormancy [118]. Further analysis [93] showed that 42.8% of DTCs from patients with no evidence of disease displayed NR2F1 upregulation, while only 10.3% DTCs from patients with metastatic diseases displayed such upregulation. Inactivation of NR2F1 was found to be linked to dormant tumour cells switching into the proliferative state, causing systemic recurrence in vivo. Furthermore, significant higher mRNA levels of TGFβ26, which was another gene relevant to bone marrow DTC dormancy [119], was detected in DTCs from patients with no evidence of disease compared with DTCs from patients with advanced diseases. Similarly, by using single-cell qPCR, Sun (2022) found DTCs from patients with advanced disease displayed a decreased level of macroH2A variants expression [94]. It was found that single dormant DTCs in head and neck squamous cell carcinoma animal models displayed an increased level of macroH2A variants expression compared to proliferative primary and metastatic lesions. These results suggested that macroH2A variant expression might be relevant in inducing dormancy in different cancers.

### 3.4. Bone Marrow Microenvironments and DTCs

A large number of studies have suggested that bone provides a nurturing microenvironment for bone marrow DTCs to survive, proliferate, escape from immune regulation, and eventually form multi-organ metastases. The bone marrow is home to different immune cells, including T cells, myeloid cells, and macrophages. Studies have shown that this immune microenvironment might be advantageous for the colonization and growth of DTCs [120,121]. Hematopoiesis occurring in the bone marrow can give rise to both erythrocytes and leukocytes [122]. Leukocytes can be categorized into myeloid and lymphoid cells, which can be further categorized into B cells and T cells [122]. A portion of T cells are actually regulatory T cells (Tregs), which play a significant role in immune suppression [123]. Specifically, FoxP3+ regulatory T cells (Treg) inhibit the activities of pro-inflammatory T cells, namely cytotoxic T cells and T helper cells by secreting anti-inflammatory cytokines like interleukin-10 (IL-10), IL-35, and TGF-β [124,125]. The decrease in pro-inflammatory cytokines can create an opportunity for dormant DTCs to escape immune surveillance. In addition, TGF-β might boost the proliferation of DTCs, forming micrometastases [126]. Patients with prostate cancer have been shown to have an increased number of Tregs within the bone [127]. Therefore, Tregs can create a protective environment that enable both dormancy and proliferation of DTCs. In addition to Tregs, myeloid-derived suppressor cells (MDSCs) also play a key role in aiding the survival of bone marrow DTCs. MDSCs is a subset of immature myeloid cells abundant in bone marrow and can suppress T cells through processes involving nitric oxide synthase (NOS2), arginase-1 (ARG1), and the generation of reactive oxygen species (ROS) [128]. Furthermore, MDSCs have the ability to produce matrix metalloproteinases (MMPs), specifically MMP-8 and MMP-9 [129]. These MMPs play a role in controlling tumour angiogenesis and facilitating tumour extravasation from the bone marrow by degrading extracellular matrix [130].

Single-cell analysis, in particular, single-cell RNA-sequencing (scRNA-seq), enables the study of the interactions between bone marrow DTCs with bone marrow individual immune cells under high resolution. Kfoury (2021) collected bone marrow samples from patients with prostate cancer and assessed immune cells, DTCs, and stromal cells in the sample by scRNA-seq [95]. All patients included had advanced prostate cancer and spinal cord compression due to bone metastasis. Metastatic tissue, bone marrow at the vertebral level with spinal cord compression, and bone marrow from a different vertebral body distant from the tumour in all patients were collected. Bone marrow samples from patients with osteoarthritis undergoing hip replacement surgery were used as control. After phenotyping and analyzing the expression profiles with scRNA-seq of all samples, it was found that all three cancer fractions showed B lymphocyte depletion when compared with control. In addition, the metastatic tissue had an increase in macrophage proportions and tumour cells compared with other samples. By comparing the expression profiles between bone marrow at the level of cord compression and controls, controls showed a significantly up-regulation in cell cycle of hematopoietic progenitor cells and down-regulation of pathways related to stress response and immune activation. Focusing on the myeloid compartment, controls contained mostly resting monocytes, while bone marrow from prostate cancer patients contained mostly proliferative monocytes. Notably, metastatic tissues contained prominent numbers of Tumour Inflammatory Monocytes (TIMs) and Tumour Associated Macrophages (TAMs), both of which have been shown to influence the suppression of anti-tumour immune responses. Detailed analysis of lymphocytes showed metastatic tissues contained significantly higher proportions of cytotoxic T lymphocytes (CTLs), T helper 1 and 17 (TH1, TH17) than other samples. Furthermore, the functional state of these T cells in metastatic tissues were altered, showing reduced cytotoxicity expression. In contrast, natural killer (NK) cells were less abundant in metastatic tissues, possibly because they failed to infiltrate into the tumour. To explain the reduced cytotoxicity in metastatic tissues, it was found that increased number of TAMs was correlated with cytotoxic T lymphocytes exhaustion. Furthermore, CCL20 was highly expressed by TAMs. CCL20 is a cytokine that bind to the CCR6 receptor, which was expressed by TH1/17 cells. These results suggested that the CCL20/CCR6 axis might be a potential therapeutic target to relieve the immunosuppression effects by tumour cells.

### 3.5. Identification of Therapeutic Targets and Risk Stratification

Single-cell analysis of DTCs is essential for understanding tumour heterogeneity and the process of tumour metastasis. By analyzing individual DTCs, this technique identifies unique genetic markers, guiding personalized treatment, aiding in biomarker discovery, and enabling patient stratification for tailored interventions. Klein et al. (2002) applied both cCGH and dot-blot hybridization on three single bone marrow DTCs from cervical, breast, and lung cancer, respectively [89]. cCGH was first used to distinguish isolated bone marrow DTCs from normal cells by comparing their genomic aberrations. Furthermore, dot-blot hybridization showed the expression of EMMPRIN protein, known to be a stimulator of the expression of matrix metalloproteinases, which degrade stromal tissue of primary tumour and facilitate tumour invasion was found in two out of three cells, while not being expressed in normal cells [131,132]. Therefore, targeting extracellular matrix metalloproteinase inducer (EMMPRIN) protein is a potential method to prevent metastasis.

Previous studies have demonstrated that DTCs from non-metastatic disease displayed fewer chromosomal aberrations than matched primary tumours and DTCs from metastatic disease [86,89]. Therefore, Schumacher (2017) hypothesized that DTCs with more copy number variations indicated more advanced disease. DTCs from bone marrow in patients with operable esophageal cancer were isolated, and patient outcome was correlated with aberrant genome per cell (PAG) values. Results showed that DTCs with PAG values above the threshold were associated with poorer survival (median survival difference = 12.5 months, *p* = 0.02) [96]. Therefore, analyzing DTCs might be a useful approach to identify patients who require intensive therapy. Another study [76] found that a single HER2-amplified DTC was strongly associated with reduced mortality, and therefore these patients might benefit from anti-HER2 treatment.

## 4. Future Perspectives

The rarity of DTCs and complex environment of bone marrow have hindered our understanding of DTCs in the past few decades. However, the emergence of single-cell analysis has contributed to solving this problem. Improvements in methods for detecting, isolating, and characterizing rare cells have facilitated progress in understanding the biology of DTCs.

Results from previous studies on bone marrow DTCs have facilitated our understanding of tumour cell dissemination and cancer progression. These include tumour heterogeneity, the time of tumour cell dissemination, tumour dormancy, and the relationship between DTCs and the bone marrow microenvironment. Although detection and genomic profiling of DTCs have contributed substantially to our knowledge of DTCs, this knowledge has not been translated into clinical use. The only clinically used treatment that target bone marrow DTCs is adjuvant bisphosphonate. It has been shown effective for eliminating bone marrow DTCs in a dormant state because of its cell-cycle-independent anti-angiogenesis and anti-cell adhesion [133,134,135]. However, owning to the heterogeneity and dormancy of DTCs, studies on treatment protocol targeting multiple pathways may be warranted.

It is widely accepted that tumour cells undergo epithelial-to-mesenchymal transition (EMT) before shedding from the primary site [136]. During the transition, tumour cells typically show loss of cell–cell adherence proteins and apical–basal polarity [137,138]. Down-regulation of epithelial markers such as EpCAM, and up-regulation of mesenchymal markers such as vimentin are also observed during the transition [138]. However, some studies have challenged the notion that EMT is necessary for tumour invasion and metastasis [139,140,141]. Indeed, bone marrow DTCs in some cancers were found to be epithelial in nature [20,142,143]. In addition, many advanced carcinomas possess the characteristics of epithelial cells, such as high levels of E-cadherin expression [144,145]. One possible explanation is that tumour cells may only undergo partial EMT and revert to epithelial morphology at the disseminated site [146,147]. Indeed, animal studies have shown that bone marrow DTCs have to revert into epithelial phenotypes before causing metastasis [148,149]. Comparison of expression profiles of DTC pools and CTCs in patients with metastatic breast cancer cells showed the upregulation of the stem cell marker, mesenchymal marker, and epithelial marker in DTC pools relative to CTCs [143]. The presence of bi-phenotypic (epithelial and mesenchymal) expressions in DTCs, but not in CTCs, suggested that epithelial–mesenchymal plasticity might be essential for tissue invasion in tumour cells. Adding complexity to the whole picture, using mouse models of breast cancer, study has shown that myeloid progenitor cells in bone marrow might play a role in stimulating mesenchymal to epithelial transition of DTCs [150]. Current methods for DTC detection usually rely on its epithelial state characterized by cytokeratin and EPCA expression might be inadequate. Furthermore, the finding of concomitant presence of DTCs with epithelial-like and mesenchymal-like states alone cannot confirm EMT unless the clonal lineage of original epithelial cells and mesenchymal cells is demonstrated. Therefore, characterizing DTC phenotypes with multi-marker detection platform followed by single-cell copy number profiling to validate the genetic lineage is important.

Previous single-cell studies on tumour dormancy of DTCs mainly focused on the molecular pathway or genetic mutations (Figure 1). However, studies have shown that the bone microenvironment also plays a role in inducing dormancy. Dormant DTCs were clustered in the hematopoietic stem-cell niche of bone marrow, in which quiescent hematopoietic stem cells are located [151,152]. Bone morphogenic protein 7 (BMP-7), which is secreted by bone stromal cells, could induce dormancy in prostate cancer cells through upregulation of p38 signalling and N-Myc downstream-regulated 1 (NDRG1), which is a metastasis suppressor gene [153]. Spatial transcriptomic profiles data using single-cell RNA sequencing can be used to evaluate the cell–cell interaction between DTCs and bone marrow cells (Figure 2).

Another unanswered question regarding to bone DTCs is their ability to cause cancer local recurrence. Animal studies have shown that circulating tumour cells are able to re-infiltrate into the tumour of origin by a process called tumour self-seeding [154]. The selection of aggressive clones during tumour self-seeding might accelerate primary tumour growth, angiogenesis, and cause local recurrence [154]. These experimental results have also been supported by clinical observations. The presence of bone marrow DTCs at diagnosis in patients with breast cancer has shown to be associated with high risk of local relapse and poor survival [20,155,156]. Nonetheless, it remains unclear what tumour-specific or DTC-specific factors can contribute to self-seedlings. Whether DTCs need to obtain additional mutations before infiltrating into the original tumour mass warrants further study. Furthermore, the difference in ability of self-seedling among DTCs in different stages and cells in established metastasis also remains unclear. Future single-cell studies comparing the molecular profiles of DTCs at diagnosis, cancer cells from local relapse tissues, and metastatic tissues might provide further information on this phenomenon.

To date, only single-cell transcriptomic and genomic analysis have been performed on bone marrow DTCs. However, to fully understand the biology of DTCs, multiple modalities are required. These include spatial transcriptomics and multi-omics study [104]. Spatial transcriptomic is a great tool to understand the crosstalk between bone marrow cells and DTCs, which might be relevant to the dormancy and immunosuppressive effects of cancer. Meanwhile, multi-omics study allows us to understand the interplay between different classes of biomolecules.

## 5. Conclusions

This comprehensive review first discusses the methodology of single-cell analysis in studying bone marrow DTCs, focusing on enrichment, detection, isolation, and characterization of individual DTCs. This review also underscores the importance of characterizing bone marrow DTCs due to their pivotal roles in bone and visceral metastasis. The molecular characterization of DTCs, facilitated by single-cell analysis, sheds light on their heterogeneity and evolutionary dynamic, challenging the traditional cancer progression models. The heterogeneity among DTCs has been highlighted, impacting the efficacy of adjuvant therapies. Furthermore, through methods like single-cell genomic and transcriptomic analysis, researchers have dissected the complex biology of DTCs, elucidating the diverse molecular pathways involved in tumour dormancy and interactions of DTCs with immune cells and stromal cells. Lastly, the identification of therapeutic targets and biomarkers through single-cell analysis offers promising avenues for personalized interventions and improved patient outcomes.

Future research directions include exploring epithelial–mesenchymal plasticity, investigating mechanisms of cancer local recurrence, and leveraging advanced modalities like spatial transcriptomics and multi-omics approaches. By embracing a multidimensional understanding of DTC biology, we can advance precision oncology and develop targeted therapies tailored to individual patients. In essence, single-cell analysis of bone marrow DTCs represents a transformative approach that not only deepens our understanding of cancer metastasis but also holds promise for translating research findings into clinical applications, ultimately enhancing cancer diagnosis, treatment, and patient care.

## Figures and Tables

**Figure 1 diagnostics-14-02172-f001:**
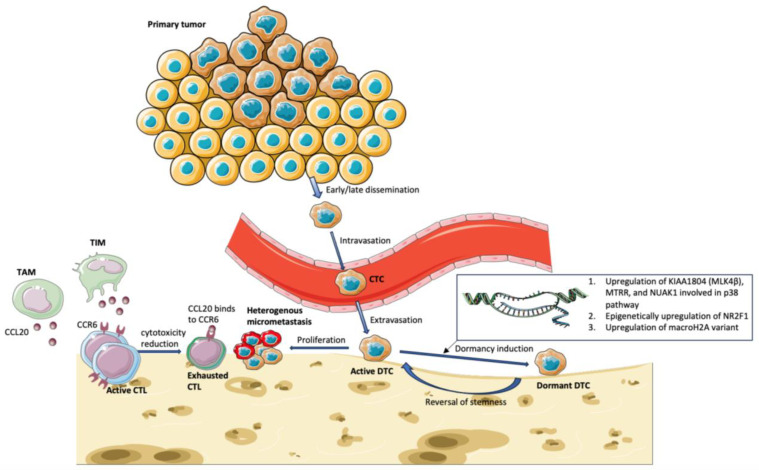
The biology of DTCs elucidated by single-cell analysis. CTC: circulating tumour cells, DTC: disseminated tumour cell, CTL: cytotoxic T lymphocyte, TIM: Tumour Inflammatory Monocyte, TAM: Tumour Associated Macrophage.

**Figure 2 diagnostics-14-02172-f002:**
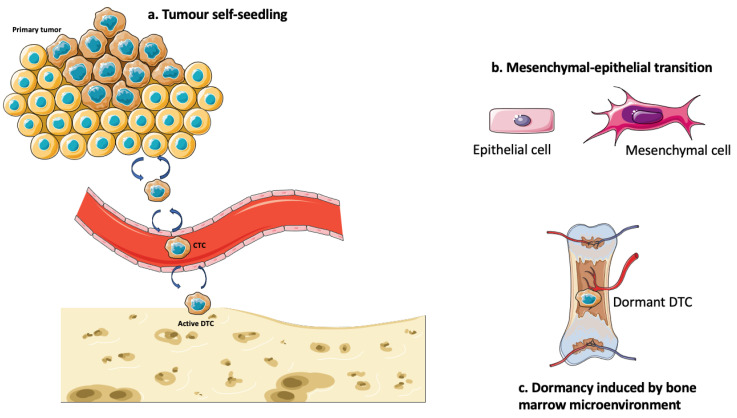
Biology of DTCs that have not been studied by single-cell analysis.

**Table 1 diagnostics-14-02172-t001:** Summary of single-cell analysis on bone marrow DTCs.

Study	Type of Cancer	Enrichment Method	Identification Method	Single-Cell Sorting	WGA/WTA	Molecular Profiling	Target Gene/Mutation
**Cancer progression model**							
Klein 1999 [30]	Cancer of unknown origin with liver metastasis	Density-gradient centrifugation (Ficoll)	Immunofluorescence staining (Cytokeratin)	Micromanipulation	Ligation-mediated PCR	cCGH, Microsatellite analysis, PCR-RFLP	Copy number aberration, LOH
Schmidt-Kittler 2003 [34]	Non-metastatic and metastatic breast cancer	Density-gradient centrifugation (Ficoll)	Immunocytochemical staining (Cytokeratin)	Micromanipulation	Ligation-mediated PCR	cCGH, Microsatellite analysis	Copy number aberration, LOH
Schardt JA 2005 [35]	Non-metastatic and metastatic breast cancer	Density-gradient centrifugation (Ficoll)	Immunocytochemical staining (Cytokeratin)	Micromanipulation	Ligation-mediated PCR	cCGH, Microsatellite analysis, PCR-RFLP, qPCR	Copy number aberration, LOH
Gangnus 2004 [32]	Non-metastatic breast cancer	N/A	Immunofluorescence staining (Cytokeratin)	Laser microdissection	Ligation-mediated PCR	cCGH	Copy number aberration
Stoecklein 2008 [85]	Non-metastatic oesophageal cancer	Density-gradient centrifugation (Ficoll)	Immunofluorescence staining (Cytokeratin)	Micromanipulation	Ligation-mediated PCR	cCGH, qPCR	Copy number aberration
Weckermann 2009 [86]	Non-metastatic and metastatic prostate cancer	Density-gradient centrifugation (Ficoll)	Immunocytochemical staining (Cytokeratin)	Micromanipulation and Laser microdissection	Ligation-mediated PCR	cCGH	Copy number aberration
Mathiesen 2011 [75]	Non-metastatic and metastatic breast cancer	Density-gradient centrifugation (Ficoll)	Immunocytochemical staining (Cytokeratin)	Micromanipulation and Laser microdissection	GenomePlex^®^ Single-Cell Whole-Genome Amplification Kit but using the Titanium Taq DNA polymerase	aCGH	Copy number aberration
Moller 2013 [74]	Non-metastatic breast cancer	Density-gradient centrifugation (Ficoll)	Immunocytochemical staining (Cytokeratin)	Micromanipulation	GenomePlex^®^ Single-Cell Whole-Genome Amplification Kit	cCGH, NGS	Copy number aberration, LOH
Demeulemeester 2016 [87]	Non-metastatic breast cancer	Density-gradient centrifugation (Ficoll)	Immunocytochemical staining (Cytokeratin)	Micromanipulation and Laser microdissection	GenomePlex^®^ Single-Cell Whole-Genome Amplification Kit but using the Titanium Taq DNA polymerase	NGS	Copy number aberration, SNP
Wu 2016 [88]	Metastatic prostate cancer	Density-gradient centrifugation (Ficoll)	Immunocytochemical staining (EpCAM)	Micromanipulation	GenomePlex^®^ Single-Cell Whole-Genome Amplification Kit	SNP-array	Somatic copy number aberration
**Heterogeneity of bone marrow DTCs**							
Klein 2002 [89]	Metastatic and non-metastatic breast, prostate, GI cancer	Density-gradient centrifugation (Ficoll)	Immunocytochemical staining (Cytokeratin)	Micromanipulation	Ligation-mediated PCR	cCGH, Single-stranded conformational polymorphism analysis,	Copy number aberration, TP53 gene screening
Czyż 2014 [90]	Metastatic breast cancer	Density-gradient centrifugation (Ficoll)	Immunocytochemical staining (Cytokeratin)	Micromanipulation	Ampli1™ Whole-Genome Amplification Kit	aCGH	Copy number aberration
**Tumour dormancy**							
Holcomb 2009 [91]	Non-metastatic and metastatic prostate cancer	N/A	Immunocytochemical staining (EpCAM)	Micromanipulation	Ligation-mediated PCR	aCGH	Copy number aberration
Chéry 2014 [92]	Non-metastatic and metastatic prostate cancer	Density-gradient centrifugation (Ficoll)/Immunomagnetic separation	Immunofluorescence staining (EpCAM)	Micromanipulation	NuGen	Expression microarray	Gene expression
Sosa MS 2015 [93]	Non-metastatic and metastatic prostate cancer	Density-gradient centrifugation (Ficoll)/Immunomagnetic separation	Immunofluorescence staining (EpCAM)	Micromanipulation	NuGen	Expression microarray	Gene expression
Sun 2022 [94]	Non-metastatic and metastatic prostate cancer	Density-gradient centrifugation (Ficoll)/Immunomagnetic separation	Immunofluorescence staining (EpCAM)	Micromanipulation	NuGen	qPCR	Gene expression
**Bone marrow microenvironment**							
Kfoury 2021 [95]	Prostate cancer with bone metastasis	N/A	Flow cytometry	Fluorescence-activated cell sorting	N/A	scRNAseq	Gene expression
**Identification of therapeutic targets and biomarkers**							
Klein 2002 [89]	Non-metastatic and metastatic cervical, lung, breast cancer	Density-gradient centrifugation (Ficoll)/Immunomagnetic separation	Immunofluorescence staining (EpCAM)	Micromanipulation	Ligation-mediated PCR	cCGH, Dot-blot hybridization	Copy number aberration, gene expression
Hoffmann 2017 [76]	Surgically treated sophageal cancer	Density-gradient centrifugation (Ficoll)	Immunofluorescence staining (Cytokeratin) (EpCAM)	Micromanipulation	Ampli1™	cCGH, qPCR	Copy number aberration, ERBB2 mutation
Schumacher 2017 [96]	Operable esophageal adenocarcinoma	Density-gradient centrifugation (Ficoll)	Immunofluorescence staining (Cytokeratin)	Micromanipulation	Ampli1™	cCGH	Copy number aberration

DTC: disseminated tumour cells, WGA: whole-genome amplification, WTA: whole-transcriptome amplification, WGS: whole-genome sequencing, cCGH: comparative genomic hybridization, qPCR: quantitative polymerase chain reaction, PCR: polymerase chain reaction, RFLP: restriction fragment length polymorphism.

## Data Availability

The datasets used and/or analyzed during the current study are available from the corresponding author on reasonable request.

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
