# Peer review of "Single-Cell Analysis of Bone-Marrow-Disseminated Tumour Cells"

_diagnostics, 2024, doi:10.3390/diagnostics14192172_

Round 1

Reviewer 1 Report

Comments and Suggestions for Authors

The manuscript by So et al summarizes the work done to identify disseminated tumor cells that reside in the bone marrow. This particular reservoir of cells may hold the key to understanding and ultimately treating metastatic disease from different cancers.

The manuscript was comprehensive and interesting.  Other than some minor editing, I found the manuscript informative and helpful.

Some minor edits are suggested:

1. The first paragraph in the INTRODUCTION section seems to be instructions for the authors. Please remove.

2. Please assure all abbreviations are accounted for. For example, the molecules mentioned in line 63 should have their abbreviations properly notated.

Author Response

Comment 1: The first paragraph in the INTRODUCTION section seems to be instructions for the authors. Please remove.

Thank you for pointing this out. We agree with this comment. Therefore, we have removed the first paragraph in the INTRODUCTION section.

Comment 2: Please assure all abbreviations are accounted for. For example, the molecules mentioned in line 63 should have their abbreviations properly notated.

Thank you for pointing this out. We agree with this comment. Therefore, we have made sure all abbreviations were probably notated. Changes include line 53 to 58 [Molecular crosstalk between tumour cells and the bone marrow micro-environment is mediated by direct cell-to-cell interactions and paracrine signalling molecules, such as transforming growth factor β (TGFβ), fibroblast growth factors (FGFs), platelet-derived growth factor (PDGF), Insulin-like growth factor (IGF), Jagged, parathyroid hormone-related protein (PTHrP), and receptor activator of nuclear factor kappa-Β ligand (RANKL)], line 135 to 136 [Currently, the detection of bone marrow DTCs is mainly based on either immunocytochemistry or reverse transcription polymerase chain reaction (RT-PCR) analysis], line 157 to 159 [include mucin-1 (MUC-1), human epidermal growth factor receptor 2 (HER2), CD 176, aldehyde dehydrogenase 1 (ALDH1), urokinase-type plasminogen activator receptor (uPAR)], line 184 to 185 [including cytokeratin-19, human mammaglobin, twist-related protein 1 (TWIST1)], line 324 to 325 [namely adenomatous polyposis coli (APC) and cadherin-1 (CDH1)], line 488 to 490 [Specifically, FoxP3+ regulatory T cells (Treg) inhibit the activities of pro-inflammatory T cells, namely cytotoxic T cells and T helper cells by secreting anti-inflammatory cytokines like interleukin-10 (IL-10), IL-35, and TGF-β (123, 124)], line 545 to 547 [Therefore, targeting extracellular matrix metalloproteinase inducer (EMMPRIN) protein is a potential method to prevent metastasis.], line 612 to 613 [could induce dormancy in prostate cancer cells through upregulation of p38 signaling and N-Myc downstream regulated 1 (NDRG1)]

Reviewer 2 Report

Comments and Suggestions for Authors

The article by Kevin WL SO et al. is a review on the topic of disseminated tumor cell (DTC) analysis in bone marrow using single-cell analysis techniques.

The importance of DTC in metastasis is discussed, particularly in bone marrow, which is a common site for metastatic growth. It is mentioned that DTC can remain dormant but can be activated and lead to metastasis.

The various technologies used to enrich, isolate and characterize DTC are reviewed.

The genomic and transcriptomic analysis methods for DTC, such as single-cell sequencing and comparative genomic hybridization, are discussed. It is mentioned that DTC exhibit genetic heterogeneity, which may influence their behavior and response to treatment.

It is highlighted that bone marrow provides a unique microenvironment conducive to DTC survival, including interactions with immune cells and stromal cells.

Overall, the review is a useful resource for understanding the current state of DTC research, but requires more critical analysis and discussion.

While reading the article, the following comments and recommendations arose:

1.The article summary should be improved. It does not reflect the main results and conclusions.

2. If the authors add more data Single-Cell Analysis of Bone Marrow Disseminated Tumour Cells, this will improve the quality of the literature review.

Author Response

1. The article summary should be improved. It does not reflect the main results and conclusions.

Thank you for pointing this out. We agree with this comment. Therefore, we have improved the part of ‘CONCLUSION’. The changes are as follow: 

This comprehensive review first discusses the methodology of single-cell analysis in studying bone marrow DTCs, focusing on enrichment, detection, isolation, and characterization of individual DTCs. The review also underscores the importance of characterizing bone marrow DTCs due to their pivotal roles in bone and visceral metastasis. The molecular characterization of DTCs, facilitated by single-cell analysis, sheds light on their heterogeneity and evolutionary dynamic, challenging the traditional cancer progression models. The heterogeneity among DTCs has been highlighted, impacting the efficacy of adjuvant therapies. Furthermore, through methods like single-cell genomic and transcriptomic analysis, researchers have dissected the complex biology of DTCs, elucidating the diverse molecular pathways involved in tumour dormancy and interactions of DTCs with immune cells and stromal cells. Lastly, the identification of therapeutic targets and biomarkers through single-cell analysis offers promising avenues for personalized interventions and improved patient outcomes.  

Future research directions include exploring epithelial-mesenchymal plasticity, investigating mechanisms of cancer local recurrence, and leveraging advanced modalities like spatial transcriptomics and multi-omics approaches. By embracing a multidimensional understanding of DTC biology, we can advance precision oncology and develop targeted therapies tailored to individual patients.  In essence, single-cell analysis of bone marrow DTCs represents a transformative approach that not only deepens our understanding of cancer metastasis but also holds promise for translating research findings into clinical applications, ultimately enhancing cancer diagnosis, treatment, and patient care.

2. If the authors add more data Single-Cell Analysis of Bone Marrow Disseminated Tumour Cells, this will improve the quality of the literature review.

Thank you for pointing this out. We agree with this comment. Therefore, we have cited all survival data in the reviewed papers (line 554 to line 555, line 366).